# Higher Level of Sports Activities Participation during Five-Year Survival Is Associated with Better Quality of Life among Chinese Breast Cancer Survivors

**DOI:** 10.3390/cancers13236056

**Published:** 2021-12-01

**Authors:** Yuan-Yuan Lei, Suzanne C. Ho, Ka-Li Cheung, Victoria A. Yeo, Roselle Lee, Carol Kwok, Ashley Cheng, Frankie K. F. Mo, Winnie Yeo

**Affiliations:** 1Department of Clinical Oncology, Prince of Wales Hospital, The Chinese University of Hong Kong, New Territories, Hong Kong, China; yuanyuanlei@cuhk.edu.hk (Y.-Y.L.); carleycheung@cuhk.edu.hk (K.L.C.); yeob2003@gmail.com (V.A.Y.); roselle_lee@hotmail.com (R.L.); frankie@clo.cuhk.edu.hk (F.K.F.M.); 2Division of Epidemiology, The Jockey Club School of Public Health and Primary Care, The Chinese University of Hong Kong, New Territories, Hong Kong, China; suzanneho@cuhk.edu.hk; 3Department of Clinical Oncology, Princess Margaret Hospital, Hong Kong, China; kwockch@hl.org.nk (C.K.); ackcheng@ha.org.hk (A.C.); 4State Key Laboratory in Oncology in South China, Faculty of Medicine, Hong Kong Cancer Institute, The Chinese University of Hong Kong, New Territories, Hong Kong, China

**Keywords:** breast cancer, sports activity, quality of life, the first five years of survival, Chinese women

## Abstract

**Simple Summary:**

Engaging in sports activities is strongly encouraged for cancer survivors. We aim to investigate the association between the level of sports activities participation post-diagnosis and quality of life (QoL) among Chinese women with early-stage breast cancer during the first five years of survival. Notably, we confirm in this work that regular participation in sports activities following cancer diagnosis may have a positive effect on patients’ QoL.

**Abstract:**

Evidence about the association between the level of sports activities participation post-diagnosis and quality of life (QoL) among Chinese women with early-stage breast cancer is limited. A validated modified Chinese Baecke questionnaire was used to prospectively measure sports activities among a breast cancer cohort at four time-points: baseline and 18, 36, and 60 months after diagnosis (sports activities during the previous 12 months before each interview); QoL was measured at the same time. In total, 1289, 1125, and 1116 patients were included in the analyses at 18-, 36- and 60-month follow-up, respectively. The percentages of patients who belonged to no (0 metabolic equivalent of task (MET)-hours per week), low-level (<10 MET-hours/week), and high-level (≥10 MET-hours/week) sports activities group were 20.7%, 45.5%, and 33.8% at 18-month follow-up, respectively; the corresponding figures at 36 and 60 months differed slightly. Using data from the three follow-ups, generalized estimating equation (GEE) analyses showed that higher levels of sports activities participation were associated with better QoL in several items as well as fewer symptoms. The present findings in Chinese women with breast cancer provide important evidence on the beneficial effect of regular participation in sports activities following cancer diagnosis on patients’ QoL.

## 1. Introduction

Survival analysis based on cancer registry data indicates that the incidence of breast cancer has increased steadily; on the other hand, in recent decades in Hong Kong, the mortality rate has only increased slightly over the same period [1]. These trends have resulted in a rising number of women who will live longer as breast cancer survivors. According to the American Cancer Society (ACS), a cancer survivor is defined as any individual who has been diagnosed with cancer [2]. As the survival rate quantitatively increases, the improvement of quality of life (QoL) becomes a critical long-term management issue for breast cancer survivors [3,4]. Previous studies have reported that both newly diagnosed breast cancer patients and long-term breast cancer survivors have worse QoL than healthy female controls [5,6]. As such, identifying effective strategies, especially by the adoption of healthy lifestyles, to improve QoL has been explored among breast cancer survivors.

Among modifiable lifestyle habits after a breast cancer diagnosis, participation in sports activities is one of the factors that has been associated with a reduced risk of mortality [7,8,9] as well as enhanced QoL [10,11,12,13]. A large body of short-term, structured exercise interventional programs has been completed by breast cancer patients. A few recently published meta-analyses have summarized the evidence and reported positive associations between exercise and QoL [14,15,16]. The above-mentioned interventional studies were mainly based on Western populations, while reports on Asian populations remain scarce. Another interesting finding was that completely or partially supervised training programs appeared to achieve greater improvement in QoL than those predominantly unsupervised or home-based ones [17]. On the other hand, observational studies evaluate patients’ self-motivated participation in sports activities and are reflective of patients’ daily habits. Up to now, several observational studies have been reported, including cohort studies and cross-sectional studies, suggesting an association between regular sports activities engagement and better QoL [18,19,20,21,22,23,24,25,26,27,28,29]. However, among these observational studies, only one cohort study has been reported in Chinese women with breast cancer [20].

Although numerous studies have reported positive associations between exercise and QoL among breast survivors, the underlying mechanisms remain unclear. The results from a lifestyle modification program, including adopting a healthy diet pattern and doing exercises, suggested that lifestyle factors may impact QoL positively, possibly through weight loss [30]. It is well recognized that Asian women have a lower body mass index (BMI) than Western women [31]. In addition, our previous study showed that less than 10% of Chinese women with breast cancer were obese, according to the BMI classification of the World Health Organization (WHO) [32]. Taken together, these data suggest that the beneficial impact of exercise on QoL may differ between Western and Chinese women. Hence, studies that specifically address this aspect in Asian patients are still essential.

The present study aims to evaluate the association between the level of sports activities and QoL among Chinese women with early-stage breast cancer during the first five years of their survival. It hypothesizes that a higher level of sports activities is associated with better QoL.

## 2. Materials and Methods

### 2.1. Patients

The Hong Kong NTEC-KWC Breast Cancer Survival Study (HKNKBCSS) was a prospective study among Chinese women with early-stage breast cancer that was initiated in 2011. It aimed to evaluate the associations between lifestyle factors, especially soy isoflavones intake, and the outcomes of breast cancer [33,34,35,36,37]. The study was approved by the Joint CUHK-NTEC Clinical Research Ethics Committee and the KWC Research Ethics Committee of the Chinese University of Hong Kong and the Hong Kong Hospital Authority.

The enrollment of this cohort took place between January 2011 and February 2014 in two regional cancer centers in Hong Kong. Consecutive breast cancer patients who attended outpatient clinics were identified for the study; eligible patients were invited to participate in the study. Inclusion criteria included Chinese females who were of any age, with histologically confirmed breast cancer at American Joint Committee on Cancer (AJCC) stage 0–III [38], mentally stable, able to read Chinese, and who did not have a prior history of breast or other cancers. These patients should have a diagnosis of breast cancer of no more than 12 months from study entry. Written informed consent was obtained from each individual prior to study entry. Each participant was interviewed at four time-points according to the study protocol: baseline at study entry, and 18, 36, and 60 months post-diagnosis.

### 2.2. Data Collection of Socio-Demographic Characteristics and Clinical Information

Each participant had a face-to-face assessment by trained interviewers at baseline and during the three follow-ups. During the assessment, patients’ socio-demographic information was collected. The clinical records were retrieved for details of breast tumor characteristics and anti-cancer treatment. In addition, disease recurrence and survival status were updated periodically by reviewing medical records during the follow-ups or by asking patients or family contacts through face-to-face interviews or telephone calls.

### 2.3. Data Collection of Lifestyle Factors

Standardized questionnaires were used to collect lifestyle factors, such as habits of dietary intake and physical activity. Anthropometric measurements included body weight and height. Body mass index (BMI) classification was based on criteria in the Asia-Pacific region, and included four groups, as follows: underweight <18.5 kg/m^2^, normal 18.5–22.9 kg/m^2^, overweight 23–24.9 kg/m^2^, and obese ≥25 kg/m^2^ [39].

### 2.4. Measurements of Sports Activity

A validated modified Chinese Baecke questionnaire (Appendix A) was used to measure the level of physical activities, which consisted of four parts: at work, doing housework, at leisure time (time excluding working and playing sports or exercise), and doing sports [40]. At baseline assessment, patients were asked to report their physical activities in the previous year before cancer diagnosis. During the 18-, 36- and 60-month follow-ups, patients recalled habitual physical activities involvement over the previous 12 months before the interviews.

Sports activities means physical activity in doing sports. We did not include other physical activities such as housework and childcare. Among the four parts of physical activities, sports activity is the one recommended by the World Cancer Research Fund/American Institute for Cancer Research (WCRF/AICR) cancer survivor guidelines and has been extensively explored by previous studies. In order to be more comparable with available evidence in the literature, the present study only analyzes data from one part of the Chinese Baecke questionnaire, namely, sports activities that contribute to overall physical activities. In the questionnaire, each individual who did sports was asked to specify the activities they were involved in categorically (up to 2 types of sports activities) and also to recall the number of hours per week and months per year they had engaged in each activity. The Ainsworth compendium of physical activity provided a MET code for each sport [41]. Based on these data, a score of metabolic equivalent of task (MET)-hours per week could be calculated by multiplying the corresponding MET value of the activity by the time (hours per week) engaged in a particular sports activity [41]. Subsequently, each individual would get a final level of sports activities by summing the score of MET-hours per week of the one or two sports activities they were engaged in.

The recommendation from the ACS and the WCRF/AICR suggests that cancer survivors should be moderately physically active, equivalent to brisk walking, for at least 30 min every day [2,42]. This recommended level of sports activity can be operationalized as being engaged in fast walking or other moderate activities for about 30 min per day for at least 5 days per week. According to the above-mentioned methods for calculation, the recommended level of sports activities is equaled to 10 MET-hours/week (calculated as follows: 4.0 METs/hour × 0.5 h/day × 5 days/week). As such, patients in the present study were divided into three groups according to their level of sports activities, as follows: no (0 MET-hours/week), low-level (<10 MET-hours/week), and high-level (≥10 MET-hours/week) sports activities. Patients in the high-level sports activities group belonged to those who met the recommendations for cancer survivors [2,42].

### 2.5. Measurement of QoL Using European Organization for Research and Treatment of Cancer Quality of Life Questionnaire Core 30 (EORTC QLQ-C30)

At each assessment, a globally accepted questionnaire, EORTC QLQ-C30, was used to measure patients’ QoL [43]. EORTC QLQ-C30 was developed to assess a range of cancer-specific QoL issues [44]. Consisting of thirty cancer-specific questions with multiple-point scales, it includes a global health status, five functional domains (physical, role, emotional, cognitive, and social), and nine symptom profiles (fatigue, nausea and vomiting, pain, dyspnea, insomnia, appetite loss, constipation, diarrhea, and financial difficulties). In the analyses, the multiple-point scale in each item would be transformed into a standard score, ranging from 0 to 100. Higher scores for global health status and the five functioning domains indicate better QoL, while lower scores for symptom items suggest fewer symptoms or problems [45].

### 2.6. Statistical Analysis

The characteristics of participants were compared among patients who engaged in different levels of sports activity using ANOVA for continuous data and the chi-squared test for categorical data. Univariate and multivariate linear regression analyses were used to investigate the association between levels of sports activities and QoL at 18-, 36- and 60-month follow-up cross-sectionally. As patients were longitudinally followed up at 18, 36, and 60 months, this study further used a generalized estimating equation (GEE) model to evaluate the longitudinal association between sports activities and QoL over the three follow-ups. Similar to repeated-measure ANOVA, the GEE model is a more powerful approach. It takes into consideration the correlations between repeated measurements within the same subjects. As such, the GEE model efficiently investigates the association between participation in sports activities and QoL in a longitudinal manner after adjustment for within-subject correlation [46]. In the analyses, each QoL item is a dependent variable as a function of time and level of sports activities as well as other covariates, which include age at follow-up, education level, household income, AJCC stage, histology, human epidermal-growth-factor receptor 2 (HER 2) status, chemotherapy, endocrine therapy, BMI at follow-up, and presence of menopausal symptoms at follow-up. All statistical analysis was performed using SPSS 26.0; *p* values <0.05 based on two-sided analysis were considered statistically significant.

## 3. Results

### 3.1. Demographic, Clinical and Lifestyle Characteristics of Patients by Level of Sports Activities at 18-, 36- and 60-Month Follow-Up

In January 2019, the 60-month follow-up assessment was completed. A total of 1462 patients completed the baseline assessment, and 1310 (89.6%), 1162 (79.5%), and 1173 (80.2%) patients completed 18-, 36- and 60-month follow-up assessment, respectively. In the present study, patients who had disease recurrence, metastasis, other primary cancers, or incomplete data on sports activities or QoL were excluded; this resulted in 1289, 1125, and 1116 patients included in the analyses at 18-, 36- and 60-month follow-up, respectively. The demographic and clinical characteristics of patients who were included in the present analysis were comparable to the whole cohort. The median intervals between the diagnosis of breast cancer and the 18-, 36-, 60-month follow-ups were 19.0 months (range: 12.1–35.8), 37.9 months (range: 30.1–41.8), and 59.7 months (range: 54.1–66.0), respectively.

At the 18-month assessment, the proportions of patients who belonged to no, low-level, and high-level sports activities groups were 20.7%, 45.5%, and 33.8%, respectively, with the mean levels of sports activities being 0, 4.2, and 23.4 MET-hours/week, respectively (Table 1). Data at the 18-month follow-up are listed in Table 1. Compared to patients who did not participate in any sports activities, those who engaged in low- or high-level sports activities were more likely to have college or above education (no vs. low-level vs. high-level sports activities group: 9.0% vs. 17.9% vs. 15.4%; *p* = 0.003). In addition, the proportions of patients who were post-menopausal at follow-up were significantly higher in those who were involved in low/high levels of sports activities (74.9% vs. 77.5% vs. 82.8%; *p* = 0.027). The histology subtypes among the three groups differed slightly; more patients who had low/high sports activities had invasive ductal carcinoma (IDC) (79.8% vs. 85.7% vs. 85.1%; *p* = 0.043). Differences were also noted in the proportion of patients having HER 2 positive tumors (21.0% vs. 28.5% vs. 26.8%; *p* < 0.001) and who received chemotherapy (70.0% vs. 78.2% vs. 77.5%; *p* = 0.027). Of note, the proportion of patients with normal BMI was much higher in patients who participated in the higher level of sports activities (41.9% vs. 42.3% vs. 53.0%; *p* = 0.001). Other demographic and clinical characteristics showed no significant difference between patients participating in different levels of sports activities.

Similar analyses were conducted at the 36-month follow-up. The proportions of patients having no, low-level, and high-level sports activities were 29.4%, 38.9%, and 31.6%, respectively. The significant differences observed above in education level (patients who had college or above: 10.6% vs. 17.1% vs. 16.6%; *p* = 0.026) and menopausal status (patients who were post-menopausal: 75.5% vs. 77.6% vs. 83.1%; *p* = 0.039) were also noted at the 36-month follow-up. In addition, patients who were involved in high-level sports activities at the 36-month follow-up tended to be older (mean age: 54.9 vs. 54.7 vs. 56.4; *p* = 0.019) and had higher household income (patients with household income ≥30,000 HKD/month: 17.2% vs. 22.8% vs. 28.1%; *p* = 0.009) than those who did not engage in any sports activities. Moreover, patients who had low-level sports activities at the 36-month follow-up were more likely to have endocrine therapy (71.0% vs. 78.8% vs. 71.3%; *p* = 0.018) and report menopausal symptoms (60.7% vs. 70.1% vs. 63.2%; *p* = 0.017) than those who did not engage in any sports activities.

At the 60-month follow-up, the proportions of patients who belonged to the no, low-level, and high-level sports activities groups were similar to those in the 36-month follow-up, with the corresponding figures being 32.8%, 36.7%, and 30.5%, respectively. When compared to patients who did not engage in any sports activity, those who were engaged in low- or high-level sports activities were more likely to have college or above education (10.7% vs. 19.0% vs. 17.4%; *p* = 0.004). In addition, the proportion of patients with normal BMI were higher in those who were engaged in low- and high-level sports activities (37.4% vs. 38.8% vs. 41.8%; *p* = 0.001). No significant differences were found regarding other characteristics.

### 3.2. Linear Regression Model to Investigate the Associations between Level of Sports Activities Participation and QoL Scores at 18-, 36- and 60-Month Follow-Up Based on Cross-Sectional Analyses

At the 18-month follow-up, 1289 patients were included in the analysis. Multivariate linear regression models with adjustment for socio-demographic, clinical, and lifestyle variables indicated that when compared to patients who had no sports activity, patients with a higher level of sports activities had significantly better QoL in global health status/QoL (*p* for trend < 0.001, Table 2), physical functioning (*p* for trend = 0.008), role functioning (*p* for trend = 0.037), and emotional functioning (*p* for trend = 0.013) as well as less symptoms in fatigue (*p* for trend < 0.001) and dyspnea (*p* for trend = 0.014).

At the 36-month follow-up, 1125 patients were included in the analysis. Similar to the associations noted at the 18-month follow-up; when compared to patients who had no sports activity, women with a higher level of sports activities reported better global health status/QoL (*p* for trend = 0.028, Table 3), physical functioning (*p* for trend = 0.003), and emotional functioning (*p* for trend = 0.006). In addition, higher participation in sports activities was significantly related to fewer symptoms of fatigue (*p* for trend < 0.001) and constipation (*p* for trend = 0.002).

A total of 1116 patients were included in the analyses at the 60-month follow-up. Similar associations were noted at this follow-up. Compared with those who had no sports activity, those who participated in a higher level of sports activities reported higher scores in global health status/QoL (*p* for trend < 0.001, Table 4) and physical functioning (*p* for trend = 0.020); on the other hand, they also had higher scores (more severe symptoms) in insomnia (*p* for trend = 0.034).

### 3.3. GEE Analyses to Investigate the Association between Level of Sports Activity and QoL Sports over the Three Follow-Ups Longitudinally

Based on the GEE analysis, after controlling for three follow-up time-points and other covariables, when compared to those who had no sports activity, the higher level of sports activities was associated with better global health status/QoL, physical functioning, role functioning, and emotional functioning (Table 5). For instance, compared to patients who did not participate in sports activity, patients who participated in low-level and high-level sports activities reported higher scores in global health status/QoL (mean difference: 0.97 and 4.53, respectively; *p* for trend < 0.0001). In terms of symptom items, the higher level of sports activities was associated with lower symptom scores (less severe symptoms) in fatigue, pain, loss of appetite, and constipation. For instance, compared to patients who did not participate in sports activity, patients who participated in low-level and high-level sports activities reported lower scores in fatigue (mean difference: −0.73 and −3.52, respectively; *p* for trend < 0.0001). In addition, interactions between the level of sports activities and follow-up time-points were also tested in the GEE models, and no significant interaction was noted (*p* for interaction >0.05 for all).

## 4. Discussion

Based on a cohort study, the present report found that regular participation in sports activities after breast cancer diagnosis was significantly associated with better general health status/QoL and physical functioning during the first five years of survival. Furthermore, the positive association between sports activities and QoL persisted over time. Using the GEE model to control for the follow-up time-points, the findings indicated that women engaged in higher-level sports activities, especially meeting the sports activities recommendation for cancer survivors (≥10 MET-hours/week), tended to have better general health status/QoL, physical functioning, role functioning, and emotional functioning as well as fewer symptoms of fatigue, pain, loss of appetite, and constipation.

This study compared the characteristics of patients who participated in different levels of sports activities. Of note, patients who attended college or above were more likely to involve in higher levels of sports activities at all three follow-ups. This phenomenon has also been reported in a previous study [20]. This study also showed that a higher proportion of women who were post-menopausal participated in higher levels of sports activities, suggesting that they might have had more spare time, which allowed them to focus more on their lifestyle habits. They may also be seeking potential maneuvers, such as sports activities, to relieve menopausal symptoms. Moreover, patients with normal BMI were associated with higher levels of sports activities participation. Patients with normal BMI might be more health-conscious; on the other hand, normal BMI could be the result of regular sports activities.

There has been a series of interventional studies investigating whether short-term, structured sports activities interventions can improve the QoL of breast cancer patients. A meta-analysis by Lahart et al. assessed the benefit of sports activities interventions after adjuvant therapy for breast cancer patients [14]; 63 trials involving 5761 women were included and well-randomized to exercise or control groups. While some studies only included aerobic sports activities or resistance training, others adopted both. The results suggest that interventions with sports activities might have a small-to-moderate benefit on overall QoL and emotional, physical, and social function as well as a reduction in anxiety [14]. Covington et al., who undertook a meta-analysis that included 31 studies, reported that community-based exercise programs are also effective for improving QoL in adult cancer survivors [15]. In another recently published systematic review by Zhang et al., which included 36 studies and 3914 participants, the type of sports activities were categorized into three modes: aerobic, resistance, and a combination of aerobic and resistance. All three modes of sports activities intervention showed a significant positive effect on QoL between groups [16].

In addition to evidence obtained from interventional studies, observational studies have also evaluated the beneficial effects of sports activities on QoL in breast cancer survivors. The Health, Eating, Activity, and Lifestyle (HEAL) study enrolled a multi-ethnic cohort of 729 early-stage breast cancer survivors in the United States (US) [21]. Participation in sports activities was evaluated for around 2.5 years after diagnosis, with QoL being assessed 6–12 months thereafter using the Medical Outcomes Study Short Form Survey (SF-36). Breast cancer survivors who partook in ≥2.5 h/week of moderate to vigorous sports activities were categorized as meeting recommendations by the American College of Sports Medicine. The results showed that women who met the recommendation had better vitality, social functioning, emotional roles, and global QoL [21]. Chen et al. investigated the association between post-diagnosis sports activities and QoL among Chinese breast cancer survivors who participated in the Shanghai Breast Cancer Survival Study (SBCSS). Participants reported their level of sports activities at 3 time-points, i.e., 6, 18, and 36 months after diagnosis [20]. The General Quality of Life Inventory-74 (GQOLI-74) was used to assess QoL at 6- and 36-month follow-up after diagnosis. Patients were categorized into three groups according to their level of sports activities: a no sports activity group, a low-level group (<8.3 MET-hours/week), and a high-level group (≥8.3 MET-hours/week) based on the US Department of Health and Human Services national recommendation for recreational physical activity [47]. At the 6-month follow-up, a higher level of sports activities was associated with better physical, psychological, and social well-being as well as total QoL. Further, the average level of sports activities over the 36 months after diagnosis was positively associated with total QoL as well as physical, psychological, and social well-being [20]. Additionally, a few cross-sectional studies have also investigated the association between sports activities and QoL among breast cancer survivors [18,19]. Blanchard et al. reported that US women who met the ACS recommendations on sports activities had better QoL than those who did not [18,19]. In another study, 3344 Chinese women with breast cancer were included [25]. Patients who were engaged in moderate-intensity sports activities for at least 30 min once every week in the past month were defined as exercisers, and the others were non-exercisers. Compared to non-exercisers, exercisers had significantly higher scores of physical, role, and emotional functioning as well as global health status; they also reported milder symptoms of nausea and vomiting, pain, dyspnea, and appetite loss [25]. More recently, a cross-sectional study among 214 breast and colorectal cancer survivors in Korea also supported the association between higher levels of sports activities and better QoL [29]. Similar to previous studies, the present study provides important data from an Asian population, supporting that regular engagement in sports activities, especially exercise that meets the recommended exercise level, was associated with better QoL in several items. Of note, the questionnaires used to measure QoL in earlier studies were not identical. Nonetheless, the significant associations between a higher level of sports activities participation and global health status, physical functioning, role functioning, and emotional functioning in the present study were consistent with those reported in the US population [21]. However, the significant association between a higher level of sports activities participation and social functioning in the US population was not observed in the present study.

For interventional studies, it is notable that the exercise engagement was supervised, encouraged, and monitored in the process of research. Compared to interventional studies, observational studies evaluated patients’ engagement in sports activities in unsupervised conditions. This has the advantage of capturing patients’ real-life participation in sports activities as part of their routine daily lifestyle. Hence, data from observational studies also provide valuable information to support the benefits of regular sports activities. The present study utilized prospective data from a cohort study and investigated the association between participation in sports activities and QoL during an important time window, namely, the first five years of survival. During this stage, breast cancer patients will adapt from an immediate to a long-term cancer survivorship. The present study shows that only about 30% of local Chinese women with breast cancer achieve the recommended level of sports activities. As such, like their Western counterparts, they should be encouraged to increase their sports activities.

There are a few possible biological mechanisms that mediate potential beneficial effects on QoL with the engagement of sports activities. These include cardiovascular and pulmonary adaptations, improved musculoskeletal strength, and maintenance of mobility and independence [48,49]. In addition, sports activities can indirectly influence psychosocial well-being via self-efficacy and health status indicators [50]. Further, a growing number of studies have shown that higher participation in sports activities is associated with better long-term outcomes in women with breast cancer [7,51,52,53,54,55,56,57]. A number of proposed mechanisms have been raised to explain the role of sports activities in modifying disease progression; these include changes in insulin-like growth factor (IGF) levels, immune regulation, and changes in metabolic hormone levels [58,59,60]. Follow-up of the cohort in the present study would provide important evidence about the potential benefit of regular engagement in sports activities on long-term outcomes among Chinese breast cancer survivors.

The present study has several strengths. Firstly, it was based on a prospective study with a relatively large sample size. During each assessment, interviewers collected detailed socio-demographic, clinical, and lifestyle factors, and potential confounders were adjusted in multivariate analyses. Secondly, this study investigated the association between participation in sports activities and QoL in a longitudinal manner at three time-points during the first five years of survival among Chinese women with breast cancer, which is an important time window for cancer survivors. In addition, the GEE model used in this study allows for the control of follow-up time-points and strengthens the reliability of the findings that regular sports activities are beneficial. Thirdly, women with breast cancer showed distinct BMI profiles as well as lifestyles from their Western counterparts. As evidence on the association between sports engagement and QoL among Asian populations is limited, this study provides a piece of important evidence for patients in our geographical region. Fourthly, data from observational studies are reflective of real-life habitual practice in our studied population. However, a few limitations exist in the present study. Firstly, while the modified Chinese Baecke questionnaire includes physical activity at work, when doing housework, at leisure time, and when doing sports, the present study only used one part of the data. Secondly, the modified Chinese Baecke questionnaire has been validated in a random sample of the Hong Kong adult population and has not been specifically validated in cancer patients. Additionally, this questionnaire was based on self-reporting, and it may have resulted in potential bias in overestimation or underestimation. Wearable instruments, such as sports bracelets, can help assess the measure of sports activities in a more objective manner in future studies.

## 5. Conclusions

The association of adopting a healthy lifestyle and improvement in QoL is an important issue in the long-term management of cancer survivors. The present findings among Chinese women after their breast cancer diagnosis provide another important piece of evidence on the beneficial effect of regular participation in sports activities on QoL. Hence, health promotion programs after a breast cancer diagnosis should aim to encourage regular participation in sports activities. In addition, continual follow-up of the studied cohort will provide information on the potential benefit of engagement in regular sports activities on the outcomes of breast cancer.

## Figures and Tables

**Table 1 cancers-13-06056-t001:** Demographic, clinical, and lifestyle characteristics of patients by level of sports activities at 18-, 36- and 60-month follow-up.

Characteristics	Level of Sports Activities at 18-Month FU,				Level of Sports Activities at 36-Month FU,				Level of Sports Activities at 60-Month FU,			
	(*n* = 1289)				(*n* = 1125)				(*n* = 1116)			
	No	Low-Level Group	High-Level Group	*p*	No	Low-Level Group	High-Level Group	*p*	No	Low-Level Group	High-Level Group	*p*
	(*n* = 267, 20.7%)	(*n* = 586, 45.5%)	(*n* = 436, 33.8%)		(*n* = 331, 29.4%)	(*n* = 438, 38.9%)	(*n* = 356, 31.6%)		(*n* = 366, 32.8%)	(*n* = 410, 36.7%)	(*n* = 340, 30.5%)	
Level of sports activity, MET-hour/week, mean ± SD	0	4.2 ± 2.8	23.4 ± 14.0		0	4.3 ± 2.8	25.8 ± 16.1		0	4.6 ± 2.8	25.3 ± 17.2	
Age at FU, mean ± SD, year	53.5 ± 9.3	53.3 ± 9.1	54.3 ± 8.6	0.24	54.9 ± 8.9	54.7 ± 9.1	56.4 ± 8.1	0.019	56.4 ± 8.8	56.7 ± 9.2	58.3 ± 8.5	0.202
Education level, %				0.003				0.026				0.004
High school or below	91	82.1	84.6		89.4	82.9	83.4		89.3	81	82.6	
College or above	9	17.9	15.4		10.6	17.1	16.6		10.7	19	17.4	
Marital status, %				0.081				0.106				0.563
Married or cohabitating	70.8	68.1	74.5		68.9	70.8	75.8		69.9	73.4	71.8	
Unmarried or divorced or widowed	29.2	31.9	25.5		31.1	29.2	24.2		30.1	26.6	28.2	
Household income, HKD/month				0.054				0.009				0.109
<15,000	54.3	44.7	44.3		51.7	43.8	43.5		50	43.9	43.2	
15,000–29,999	28.5	31.2	32.1		31.1	33.3	28.4		31.7	30.7	31.5	
≥30,000	17.2	24.1	23.6		17.2	22.8	28.1		18.3	25.4	25.3	
Employment status				0.086				0.133				0.225
Full time	34.1	35.7	40.8		34.4	36.1	42.4		36.1	33.9	42.1	
Part time	16.1	12.1	14.7		14.5	11.4	12.4		12.8	13.7	11.2	
Not working	49.8	52.2	44.5		51.1	52.5	45.2		51.1	52.4	46.8	
No. of comorbidities, %				0.273				0.528				0.623
0	55.4	61.9	62.6		59.2	62.8	61.5		60.7	62.7	61.5	
1	28.5	25.3	25.9		25.7	24	27.2		24.6	26.3	25	
≥2	16.1	12.8	11.5		15.1	13.2	11.2		14.8	11	13.5	
Menopausal status at FU, %				0.027				0.039				0.256
Pre-menopausal	25.1	22.5	17.2		24.5	22.4	16.9		17.5	18	13.8	
Post-menopausal	74.9	77.5	82.8		75.5	77.6	83.1		82.5	82	86.2	
AJCC stage, %				0.518				0.472				0.051
0-I	36.3	35	36.9		35	38.6	35.1		38.8	37.3	38.2	
II	43.1	45.7	46.8		46.5	46.1	48.9		41.5	48	48.8	
III	19.9	18.6	16.3		17.5	15.3	15.4		18.3	14.6	12.4	
Missing	0.7	0.7	0		0.9	0	0.6		1.4	0	0.6	
Histology, %				0.043				0.591				0.787
IDC	79.8	85.7	85.1		84.3	83.1	84.3		83.9	84.4	83.2	
ILC	1.9	2.7	3.9		1.5	3.7	3.7		1.9	3.4	3.2	
DCIS	9.7	4.9	5		7.3	6.2	5.3		7.1	5.6	5.6	
Others	8.6	6.7	6		6.9	7.1	6.7		7.1	6.6	7.9	
ER status, %				0.267				0.993				0.662
Positive	73.1	73.1	73.4		72.8	72.8	72.5		75.1	71.2	73.8	
Negative	23.2	23.5	25.2		24.2	24.2	25		21.3	25.6	23.5	
Missing	3.7	3.4	1.4		3	3	2.5		3.6	3.2	2.6	
PR status, %				0.05				0.838				0.924
Positive	59.2	55.8	54.8		55.3	57.5	42.4		56	55.6	55.9	
Negative	37.1	39.9	43.8		41.1	39.3	55.1		39.9	41.2	41.2	
Missing	3.7	4.3	1.4		3.6	3.2	2.5		4.1	3.2	2.9	
HER 2 status, %				<0.001				0.599				0.081
Positive	21	28.5	26.8		29	27.2	24.4		24.9	27.8	23.8	
Negative	66.3	66.9	68.1		63.7	66.4	69.7		65.6	67.3	70	
Missing	12.7	4.6	5		7.3	6.4	5.9		9.6	4.9	6.2	
Type of surgery				0.403				0.904				0.517
Mastectomy	65.9	61.6	61.2		62.2	61	60.7		62	61	57.9	
Conservation	34.1	38.4	38.8		37.8	39	39.3		38	39	42.1	
Chemotherapy, %				0.027				0.826				0.601
Yes	70	78.2	77.5		75.5	76.5	77.5		72.4	75.4	75	
No	30	21.8	22.5		24.5	23.5	22.5		24.6	24.6	25	
Radiotherapy, %				0.181				0.47				0.815
Yes	67.4	73.4	70.2		69.2	72.8	71.3		69.9	69	71.2	
No	32.6	26.6	29.8		30.8	27.2	28.7		30.1	31	28.8	
Endocrine therapy, %				0.227				0.018				0.788
Yes	76.4	72.2	76.4		71	78.8	71.3		75.4	74.9	73.2	
No	23.6	27.8	23.6		29	21.1	28.7		24.6	25.1	26.8	
BMI at FU, kg/m^2^				0.001				0.153				0.001
Underweight (<18.5)	5.2	6.3	3.9		3.6	4.8	4.5		3	6.3	3.8	
Normal (18.5–22.9)	41.9	42.3	53		41.1	40.9	46.3		37.4	38.8	41.8	
Overweight (23–24.9)	19.9	20.5	22		19.3	22.4	23.3		16.9	23.9	25	
Obese (≥25)	33	30.9	21.1		36	32	25.8		42.6	31	29.4	
Presence of menopausal symptoms, %				0.677				0.017				0.313
Yes	64	65.7	63.1		60.7	70.1	63.2		62.8	66.6	61.5	
No	36	34.3	36.9		39.3	29.9	36.8		37.2	33.4	38.5	

Abbreviations: FU, follow-up; SD, standard deviation; HKD, Hong Kong dollars; AJCC, American Joint Committee on Cancer; IDC, invasive ductal carcinoma; ILC, invasive lobular carcinoma; DCIS, ductal carcinoma in situ; ER, estrogen receptor; PR, progesterone receptor; HER 2, human epidermal-growth-factor receptor 2; BMI, body mass index; MET, metabolic equivalent of task.

**Table 2 cancers-13-06056-t002:** Univariate and multivariate linear regression analyses to investigate the associations of the level of sports activities and QoL scores at 18-month follow-up cross-sectionally (*n* = 1289).

EORTC QLQ-C30	Crude					Adjusted				
	Low-Level Group (*n* = 438)		High-Level Group (*n* = 356)		*p* for Trend	Low-Level Group (*n* = 438)		High-Level Group (*n* = 356)		*p* for Trend
	Mean Difference	95% CI	Mean Difference	95% CI		Mean Difference	95% CI	Mean Difference	95% CI	
Global Health status/QoL	1.43	1.08 to 3.95	5.44	2.79 to 8.09	<0.001	1.32	1.06 to 3.69	5.07	2.56 to 7.57	<0.001
Functioning										
Physical Functioning	−0.31	−1.98 to 1.37	2.21	0.46 to 3.97	0.004	−0.38	−1.93 to 1.16	1.86	0.23 to 3.50	0.008
Role Functioning	−0.32	−2.43 to 1.78	1.71	−0.50 to 3.93	0.076	−0.05	−2.11 to 2.00	2.02	−0.15 to 4.19	0.037
Emotional Functioning	2.41	−0.27 to 5.10	3.22	0.39 to 6.05	0.033	2.97	0.57 to 5.38	3.48	0.94 to 6.02	0.013
Cognitive Functioning	0.18	−2.56 to 2.92	1.06	−1.82 to 3.94	0.432	0.14	−2.48 to 2.75	0.81	−1.96 to 3.57	0.531
Social Functioning	0.41	−1.79 to 2.61	0.44	−1.87 to 2.76	0.730	0.70	−1.45 to 2.86	0.40	−1.87 to 2.68	0.805
Symptoms										
Fatigue	−0.56	−3.14 to 2.30	−4.79	−7.80 to −1.79	<0.001	−0.90	−3.53 to 1.73	−4.31	−7.09 to −1.54	<0.001
Nausea and vomiting	−0.15	−1.07 to 0.78	−0.54	−1.52 to 0.43	0.242	−0.10	−1.03 to 0.84	−0.30	−1.28 to 0.69	0.530
Pain	1.19	−2.12 to 4.50	−3.59	−7.07 to −0.11	0.015	1.58	−1.44 to 4.61	−2.36	−5.56 to 0.83	0.060
Dyspnea	−0.60	−3.05 to 1.85	−3.41	−5.99 to −0.83	0.005	−0.88	−3.28 to 1.52	−2.98	−5.51 to −0.48	0.014
Insomnia	3.41	−0.71 to 7.52	0.79	−3.54 to 5.13	0.958	3.05	−0.84 to 6.94	0.57	−3.55 to 4.68	0.962
Loss of appetite	−0.83	−2.76 to 1.10	−2.05	−4.08 to −0.02	0.040	−0.46	−2.39 to 1.47	−1.42	−3.47 to 0.62	0.148
Constipation	−1.94	−4.81 to 0.92	−2.58	−5.59 to 0.44	0.109	−1.82	−4.72 to 1.09	−2.54	−5.61 to 0.54	0.119
Diarrhea	1.57	−0.22 to 3.35	−0.51	−2.39 to 1.37	0.329	1.15	−0.64 to 2.94	−0.66	−2.56 to 1.23	0.286
Financial impact	0.35	−3.05 to 3.75	0.14	−3.44 to 3.71	0.968	1.28	−2.01 to 4.57	1.67	−1.81 to 5.15	0.372

Univariate and multivariate linear regression models were used to investigate the associations between the level of sports activities and QoL scores. The adjusted variables included age at follow-up, education level, household income, menopausal status at follow-up, total number of comorbidities, AJCC stage, histology, HER 2 status, chemotherapy, endocrine therapy, BMI at follow-up, and presence of menopausal symptoms at follow-up. Patients in the no sports activity group were treated as the reference in calculating the mean difference of QoL. Abbreviations: 95% CI, 95% confidence intervals; EORTC QLQ-C30, European Organization for Research and Treatment of Cancer Quality of Life Questionnaire Core 30; QoL, quality of life; AJCC, American Joint Committee on Cancer; HER 2, human epidermal-growth-factor receptor 2; BMI, body mass index.

**Table 3 cancers-13-06056-t003:** Univariate and multivariate linear regression analyses to investigate the associations of the level of sports activities and QoL scores at 36-month follow-up cross-sectionally (*n* = 1125).

EORTC QLQ-C30	Crude					Adjusted				
	Low-Level Group (*n* = 586)		High-Level Group (*n* = 436)		*p* for Trend	Low-level Group (*n* = 586)		High-Level Group (*n* = 436)		*p* for Trend
	Mean Difference	95% CI	Mean Difference	95% CI		Mean Difference	95% CI	Mean Difference	95% CI	
Global Health status/QoL	−0.59	−3.20 to 2.02	3.17	0.43 to 5.91	0.021	−0.05	−2.50 to 2.40	2.85	0.27 to 5.43	0.028
Functioning										
Physical Functioning	−0.61	−2.40 to 1.18	3.05	1.17 to 4.93	0.001	−0.39	−2.08 to 1.30	2.67	0.90 to 4.45	0.003
Role Functioning	−0.32	−2.52 to 1.87	2.08	−0.23 to 4.38	0.072	0.28	−1.88 to 2.44	1.97	−0.31 to 4.24	0.087
Emotional Functioning	−0.16	−2.78 to 2.47	3.11	0.36 to 5.86	0.002	1.60	−0.75 to 3.95	3.46	0.99 to 5.93	0.006
Cognitive Functioning	−0.43	−3.22 to 2.36	0.76	−2.16 to 3.69	0.598	0.69	−1.88 to 3.26	0.46	−2.25 to 3.17	0.748
Social Functioning	−0.65	−2.71 to 1.42	1.68	−0.49 to 3.84	0.119	−0.23	−2.28 to 1.82	1.37	−0.79 to 3.52	0.205
Symptoms										
Fatigue	0.05	−2.94 to 3.04	−5.19	−8.32 to −2.05	0.001	−1.24	−3.91 to 1.44	−5.05	−7.86 to −2.23	<0.001
Nausea and vomiting	0.52	−0.86 to 1.89	−0.38	−1.82 to 1.06	0.587	0.42	−0.96 to 1.80	−0.24	−1.70 to 1.21	0.726
Pain	1.08	−2.04 to 4.20	−3.48	−6.75 to −0.21	0.033	0.66	−2.26 to 3.57	−2.86	−5.93 to 0.22	0.063
Dyspnea	0.83	−1.92 to 3.57	−0.46	−3.34 to 2.42	0.739	0.43	−2.21 to 3.07	0.22	−2.56 to 3.00	0.881
Insomnia	0.99	−3.20 to 5.18	−2.05	−6.44 to 2.34	0.347	−0.85	−4.77 to 3.07	−2.60	−6.73 to 1.53	0.214
Loss of appetite	0.23	−1.94 to 2.39	−1.86	−4.14 to 0.41	0.102	−0.09	−2.26 to 2.07	−1.83	−4.11 to 0.44	0.109
Constipation	−3.06	−6.15 to 0.04	−5.25	−8.49 to −2.00	0.002	−3.75	−6.83 to −0.68	−5.13	−8.34 to −1.88	0.002
Diarrhea	0.23	−1.91 to 2.37	−1.02	−3.26 to 1.22	0.362	−0.19	−2.33 to 1.94	−1.15	−3.41 to 1.10	0.309
Financial impact	2.12	−0.77 to 5.00	0.55	−2.47 to 3.57	0.746	2.34	−0.44 to 5.12	1.43	−1.50 to 4.37	0.356

Univariate and multivariate linear regression models were used to investigate the associations between the level of sports activities and QoL scores. The adjusted variables included age at follow-up, education level, household income, menopausal status at follow-up, total number of comorbidities, AJCC stage, histology, HER 2 status, chemotherapy, endocrine therapy, BMI at follow-up, and presence of menopausal symptoms at follow-up. Patients in the no sports activity group were treated as the reference in calculating the mean difference of QoL. Abbreviations: 95% CI, 95% confidence intervals; EORTC QLQ-C30, European Organization for Research and Treatment of Cancer Quality of Life Questionnaire Core 30; QoL, quality of life; AJCC, American Joint Committee on Cancer; HER 2, human epidermal-growth-factor receptor 2; BMI, body mass index.

**Table 4 cancers-13-06056-t004:** Univariate and multivariate linear regression analyses to investigate the associations of the level of sports activities and QoL scores at 60-month follow-up cross-sectionally (*n* = 1116).

EORTC QLQ-C30	Crude					Adjusted				
	Low-Level Group (*n* = 410)		High-Level Group (*n* = 340)		*p* for Trend	Low-Level Group (*n* = 410)		High-Level Group (*n* = 340)		*p* for Trend
	Mean Difference	95% CI	Mean Difference	95% CI		Mean Difference	95% CI	Mean Difference	95% CI	
Global Health status/QoL	1.13	−1.35 to 3.61	6.20	3.60 to 8.80	<0.001	1.43	−0.92 to 3.78	5.74	3.26 to 8.21	<0.001
Functioning										
Physical Functioning	−0.13	−1.90 to 1.63	2.58	0.73 to 4.43	0.007	−0.22	−1.87 to 1.43	2.07	0.33 to 3.81	0.020
Role Functioning	0.49	−1.65 to 2.64	0.60	−1.65 to 2.85	0.597	0.97	−1.18 to 3.12	0.44	−1.82 to 2.71	0.697
Emotional Functioning	0.06	−2.45 to 2.56	1.92	0.71 to 4.54	0.157	0.62	−1.66 to 2.91	1.40	−1.01 to 3.80	0.253
Cognitive Functioning	−0.34	−3.08 to 2.40	0.66	−2.21 to 3.53	0.661	−0.04	−2.61 to 2.52	−0.05	−2.75 to 2.65	0.971
Social Functioning	−0.53	−2.43 to 1.36	−0.18	−2.16 to 1.81	0.852	−0.53	−2.39 to 1.34	−0.95	−2.92 to 1.02	0.342
Symptoms										
Fatigue	0.55	−2.25 to 3.34	−1.79	−4.72 to 1.13	0.240	−0.03	−2.56 to 2.50	−0.95	−3.62 to 1.71	0.483
Nausea and vomiting	0.27	−0.90 to 1.43	−0.26	−1.48 to 0.96	0.692	0.37	−0.81 to 1.55	0.09	−1.15 to 1.33	0.886
Pain	1.06	−2.08 to 4.21	−0.98	−4.27 to 2.32	0.577	1.22	−1.69 to 4.14	0.09	−2.98 to 3.17	0.949
Dyspnea	−0.26	−2.79 to 2.27	−1.14	−3.79 to 1.51	0.402	−0.13	−2.59 to 2.33	0.10	−2.49 to 2.70	0.938
Insomnia	4.56	0.45 to 8.67	3.69	−0.61 to 8.00	0.086	4.07	0.20 to 7.94	4.40	0.32 to 8.48	0.034
Loss of appetite	0.42	−1.49 to 2.34	−2.19	−4.20 to −0.19	0.036	0.43	−1.47 to 2.33	−1.62	−3.62 to 0.38	0.115
Constipation	−0.93	−4.07 to 2.22	−3.54	−6.84 to −0.25	0.036	−1.63	−4.79 to 1.53	−3.32	−6.64 to 0.01	0.050
Diarrhea	0.22	−1.88 to 2.32	−1.38	−3.58 to 0.81	0.225	0.23	−1.88 to 2.34	−0.85	−3.08 to 1.37	0.455
Financial impact	−0.31	−2.99 to 2.37	−1.38	−4.19 to 1.43	0.340	0.14	−2.48 to 2.75	0.01	−2.75 to 2.76	0.997

Univariate and multivariate linear regression models were used to investigate the associations between the level of sports activities and QoL scores. The adjusted variables included age at follow-up, education level, household income, menopausal status at follow-up, total number of comorbidities, AJCC stage, histology, HER 2 status, chemotherapy, endocrine therapy, BMI at follow-up, and presence of menopausal symptoms at follow-up. Patients in the no sports activity group were treated as the reference in calculating the mean difference of QoL. Abbreviations: 95% CI, 95% confidence intervals; EORTC QLQ-C30, European Organization for Research and Treatment of Cancer Quality of Life Questionnaire Core 30; QoL, quality of life; AJCC, American Joint Committee on Cancer; HER 2, human epidermal-growth-factor receptor 2; BMI, body mass index.

**Table 5 cancers-13-06056-t005:** Univariate and multivariate linear regression analyses to investigate the associations of the level of sports activities and QoL scores at 60-month follow-up cross-sectionally (*n* = 1116).

EORTC QLQ-C30	Model 1					Model 2				
	Low-Level Group (*n* = 410)		High-Level Group (*n* = 340)		*p* for Trend	Low-Level Group (*n* = 410)		High-Level Group (*n* = 340)		*p* for Trend
	Mean Difference	95% CI	Mean Difference	95% CI		Mean Difference	95% CI	Mean Difference	95% CI	
Global Health status/QoL	0.67	−1.0 to 2.34	4.9	3.03 to 6.80	<0.001	0.97	−0.51 to 2.45	4.53	2.85 to 6.20	<0.001
Functioning										
Physical Functioning	−0.33	−1.52 to 0.86	2.61	1.38 to 3.83	<0.001	−0.27	−1.35 to 0.80	2.25	1.16 to 3.34	<0.001
Role Functioning	−0.09	−1.50 to 1.32	1.49	−0.01 to 3.00	0.042	0.29	−1.09 to 1.66	1.53	0.09 to 2.97	0.030
Emotional Functioning	0.78	−0.98 to 2.53	1.92	0.68 to 4.62	0.007	1.77	0.27 to 3.28	2.75	1.10 to 4.40	0.001
Cognitive Functioning	−0.20	−2.06 to 1.66	0.88	−1.27 to 2.88	0.419	0.26	−1.42 to 1.94	0.54	−1.31 to 2.38	0.567
Social Functioning	−0.24	−1.52 to 1.04	0.60	−0.85 to 2.05	0.389	−0.02	−1.23 to 1.19	0.30	−1.05 to 1.66	0.644
Symptoms										
Fatigue	0.08	−1.90 to 2.07	−3.90	−6.03 to −1.77	<0.001	−0.73	−2.42 to 0.97	−3.52	−5.28 to −1.75	<0.001
Nausea and vomiting	0.21	−0.49 to 0.91	−0.37	−1.08 to 0.33	0.261	0.22	−0.48 to 0.92	−0.14	−0.82 to 0.54	0.631
Pain	1.19	−0.89 to 3.28	−2.71	−4.94 to −0.48	0.011	1.17	−0.66 to 3.00	−1.74	−3.67 to 0.19	0.011
Dyspnea	0.08	−1.57 to 1.73	−1.68	−3.43 to 0.07	0.049	−0.05	−1.59 to 1.50	−0.91	−2.55 to 0.73	0.051
Insomnia	3.05	0.37 to 5.72	0.82	−2.16 to 3.80	0.682	2.06	−0.32 to 4.44	0.64	−1.99 to 3.27	0.713
Loss of appetite	−0.08	−1.37 to 1.20	−1.98	−3.22 to −0.75	0.001	−0.13	−1.38 to 1.12	−1.62	−2.83 to −0.40	0.006
Constipation	−2.04	−4.14 to 0.06	−3.74	−6.02 to −1.47	0.001	−2.45	−4.49 to −0.42	−3.64	−5.83 to −1.44	0.001
Diarrhea	0.66	−0.57 to 1.90	−1.02	−2.26 to 0.21	0.078	0.44	−0.77 to 1.64	−0.98	−2.19 to 0.22	0.082
Financial impact	0.67	−1.28 to 2.62	−0.22	−2.33 to 1.90	0.798	1.17	−0.67 to 3.02	1.09	−0.88 to 3.06	0.302

Generalized estimating equations were used to investigate the associations between the level of physical activity and QoL scores. Model 1, adjusted for follow-up time-points; Model 2, further adjusted for age at follow-up, education level, household income, menopausal status at follow-up, total number of comorbidities, AJCC stage, histology, HER 2 status, chemotherapy, endocrine therapy, BMI at follow-up, and presence of menopausal symptoms at follow-up. Patients in the no sports activity group were treated as the reference in calculating the mean difference of QoL. Abbreviations: 95% CI, 95% confidence intervals; EORTC QLQ-C30, European Organization for Research and Treatment of Cancer Quality of Life Questionnaire Core 30; QoL, quality of life; AJCC, American Joint Committee on Cancer; HER 2, human epidermal-growth-factor receptor 2; BMI, body mass index.

## Data Availability

Data sharing is not applicable to this article.

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
