# Peer review of "Higher Level of Sports Activities Participation during Five-Year Survival Is Associated with Better Quality of Life among Chinese Breast Cancer Survivors"

_cancers, 2021, doi:10.3390/cancers13236056_

Round 1

Reviewer 1 Report

This is an interesting paper on the relationship between sports activity levels and QoL in Chinese women with early-stage breast cancer during the first five years of their survival. The study is clinically relevant, the conclusions were based on a large group of participants studied at three time points. The results presented provide evidence that regular participation in sports activities is significantly associated with better overall health of patients, better QoL and physical functioning, and most importantly, the positive association between sports activity and QoL is maintained over time. This is good news not only for physicians but also for patients, further strong evidence for the role of physical activity in the health management of oncology patients.

Author Response

Response: We thank the Reviewer for these comments.

Reviewer 2 Report

Introduction:

While authors’ state only one reported study that had only Chinese women has been previously documented.  Can you give more information about why we would expect this association to be different in the population? What gap are you filling?

Methods/Results:

Given that this is longitudinal data it may be more informative to focus on chance over time instead of treating each time point separately (Results 3.2 and 3.3). That is what are the characteristics associated with increasing activity or decreasing activity; especially since it seems like at the different time point’s different association are seen.

Results: could you give the proportion or number of patients in the BMI categories?

Discussion: it seems like this study is not all that informative given that there are already interventions that have been designed and studied to improve activity level in order to improve QoL in breast cancer patients.

Overall: It is not clear to me what this study adds to the evidence base around breast cancer survivors’ physical activity and QoL. 

Author Response

Response to Reviewer 2 Comments

Point 1: 
Introduction:

While authors’ state only one reported study that had only Chinese women has been previously documented. Can you give more information about why we would expect this association to be different in the population? What gap are you filling?

Response 1: We thank the Reviewer for these comments. We have added the following description in the Introduction Section.

“Although numerous studies have reported positive associations between exercise and QoL among breast survivors, the underlying mechanisms remain unclear. Result from a lifestyle modification program, including adopting healthy diet pattern and doing exercises, suggested that lifestyle factors may impact positively on QoL possibly through weight loss.[30] It is well recognized that Asian women have lower body mass index (BMI) than Western women.[31] In addition, our previous study showed that less than 10% of Chinese women with breast cancer were obese according to the BMI classification of World Health Organization (WHO).[32] Taken together, these data suggested that the beneficial impact of exercise on QoL may differ between Western and Chinese women. Hence, studies that specifically address this aspect in Asian patients are still essential.” Please refer to Introduction section, Page 2, Line 69-78.

Reference

  1. Montagnese, C.; Porciello, G.; Vitale, S.; Palumbo, E.; Crispo, A.; Grimaldi, M.; Calabrese, I.; Pica, R.; Prete, M.; Falzone, L., et al. Quality of Life in Women Diagnosed with Breast Cancer after a 12-Month Treatment of Lifestyle Modifications. Nutrients 2020, 13, doi:10.3390/nu13010136.
  2. Chen, M.J.; Wu, W.Y.Y.; Yen, A.M.F.; Fann, J.C.Y.; Chen, S.L.S.; Chiu, S.Y.H.; Chen, H.H.; Chiou, S.T. Body mass index and breast cancer: analysis of a nation-wide population-based prospective cohort study on 1 393 985 Taiwanese women. International journal of obesity (2005) 2016, 40, 524-530, doi:10.1038/ijo.2015.205.
  3. Lei, Y.-Y.; Ho, S.C.; Kwok, C.; Cheng, A.; Cheung, K.L.; Lee, R.; Mo, F.; Yeo, W. Weight and waist-to-hip ratio change pattern during the first five years of survival: data from a longitudinal observational Chinese breast cancer cohort. BMC cancer 2021, 21, 839-839, doi:10.1186/s12885-021-08554-5.

Point 2:

Methods/Results:

Given that this is longitudinal data it may be more informative to focus on change over time instead of treating each time point separately (Results 3.2 and 3.3). That is what are the characteristics associated with increasing activity or decreasing activity; especially since it seems like at the different time point’s different association are seen.

Response 2: We thank the Reviewer for these comments. In Results 3.2, the associations between different level of participation in sports activities and QoL scores were investigated cross-sectionally. We agreed with the reviewer that the study was conducted with projected follow-up, and as such, it may be more informative to focus on change over time instead of treating each time point separately. To fully assess the longitudinal data, we applied Generalized Estimating Equations (GEE) for analysis (Result 3.3). One advantage of GEE is that it takes into consideration the correlations between repeated measurements within the same subjects (namely the changes over time). In the present study (Table 5, model 2), the within-subject correlation has been adjusted and GEE models have effectively investigated the association between participation of sports activities and QoL in a longitudinal manner. We have added more descriptions in the revised manuscript as follow.

“Similar to repeated-measure ANOVA, the GEE model is a more powerful approach. It could take into consideration the correlations between repeated measurements within the same subjects. As such, the GEE model could efficiently investigate the association between participation of sports activities and QoL in a longitudinal manner, after adjustment for within-subject correlation.[46]” Please refer to Methods section, Page 4, Line 169-173.

Reference

  1. Raboud, J.M.; Singer, J.; Thorne, A.; Schechter, M.T.; Shafran, S.D. Estimating the effect of treatment on quality of life in the presence of missing data due to drop-out and death. Quality of life research : an international journal of quality of life aspects of treatment, care and rehabilitation 1998, 7, 487-494, doi:10.1023/a:1008870223350.

Point 3: Results: could you give the proportion or number of patients in the BMI categories?

Response 3: We thank the Reviewer for these comments. The proportion of patients with different BMI categories were listed in Table 1.

Point 4: Discussion: it seems like this study is not all that informative given that there are already interventions that have been designed and studied to improve activity level in order to improve QoL in breast cancer patients.

Response 4: We thank the Reviewer for these comments. We fully agree with the Reviewer that interventional studies have provided strong evidence on the association between improved sports activities level and better QoL in breast cancer patients. For interventional studies, it is notable that the exercise engagement was supervised, encouraged, and monitored in the process of research. Compared to interventional studies, observational studies evaluated patients’ engagement in sports activities in unsupervised conditions, which has the advantage of capturing patients’ real-life participation in sports activities as part of their routine daily lifestyle. Hence, data from observational studies also provide valuable information to support the importance of participating in regular sports activities. In addition, the present study showed that only about 30% of our studied Chinese women with breast cancer have achieved the recommended level of sports activities. As such, data from the present study could help encourage participation of sports activities among Asian patients with breast cancer. We have added this description in the discussion section.

“For interventional studies, it is notable that the exercise engagement was supervised, encouraged, and monitored in the process of research. Compared to interventional studies, observational studies evaluated patients’ engagement in sports activities in unsupervised conditions. This has the advantage of capturing patients’ real-life participation in sports activities as part of their routine daily lifestyle. Hence, data from observational studies also provides valuable information to support the benefits of regular sports activities. The present study utilized prospective data from a cohort study and investigated the association between participation of sports activities and QoL during an important time-window, namely the first five-years of survival. During this stage, breast cancer patients would adapt from an immediate to a long-term cancer survivorship. In the present study, it showed that only about 30% of local Chinese women with breast cancer have achieved the recommended level of sports activities. As such, like their Western counterparts, they should be encouraged to increase their sports activities.” Please refer to Discussion section, Line 405-416.

Point 5: Overall: It is not clear to me what this study adds to the evidence base around breast cancer survivors’ physical activity and QoL.

Response 5: We thank the Reviewer for these comments. The present study has a few strengths as listed below. We sincerely hope that these strengths could fulfill the requirements to make the manuscript acceptable for publication.

“The present study has several strengths. Firstly, it was based on a prospective study with relatively large sample size. During each assessment, interviewers collected detailed socio-demographic, clinical and lifestyle factors, and potential confounders have been adjusted in multivariate analyses. Secondly, this study investigated the association between participation of sports activities and QoL in a longitudinal manner at three time-points during the first five-years of survival among Chinese women with breast cancer, which was an important time window for cancer survivors. In addition, the GEE model used in this study allows the control for follow-up time-points, and strengthen the reliability of the findings that regular sports activities is beneficial. Thirdly, women with breast cancer showed distinct BMI profile as well as lifestyles from their Western counterparts. As evidence on the association between sports engagement and QoL among Asian population is limited, this study provided a piece of important evidence for patients in our geographical region. Fourthly, data from observational studies is reflective of real-life habitual practice in our studied population.” Please refer to Discussion section, Line 431-444.

Reviewer 3 Report

The study is an important one considering the level of morbidity and mortality rate from breast cancer. However, before this study could be considered for publication, the following issues regarding the manuscript have to be addressed.

Introduction

The introduction should be implemented with most recent studies, to better understand the relation between physical activity and QoL in breast cancer survivors. Considering that the novelty of the study lies in the fact that there are few research that have studied the relationship between QOL and physical activity in Chinese breast cancer women, it would be interesting to know il the authors expect different result from the one already known for western breast cancer survivors.

Results

3.1. Demographic, clinical and lifestyle characteristics of patients by level of sports activities at  18-, 36- and 60-month follow-up

This section should be rewritten as the result’s section reported below, reporting the specific data (mean value, p) without discussion.

Discussion

Please comment these results of the 3.1 section.

 Please evidence if your data differ from those reported in manuscripts on western breast cancer survivors.

References

Please update this reference:

World Cancer Research Fund/American Institute for Cancer Research. Food, Nutrition, Physical Activity, and the Prevention 494 of Cancer: A Global Perspective. Washington, DC: AICR, 2007.

Author Response

Response to Reviewer 3 Comments

Comments and Suggestions for Authors

The study is an important one considering the level of morbidity and mortality rate from breast cancer. However, before this study could be considered for publication, the following issues regarding the manuscript have to be addressed.

Point 1: 

Introduction

The introduction should be implemented with most recent studies, to better understand the relation between physical activity and QoL in breast cancer survivors. Considering that the novelty of the study lies in the fact that there are few research that have studied the relationship between QOL and physical activity in Chinese breast cancer women, it would be interesting to know if the authors expect different result from the one already known for western breast cancer survivors.

Response 1: We thank the Reviewer for these comments. We have inserted several recent published studies in the introduction section, to better understand the relation between physical activity and QoL in breast cancer survivors.

“Among modifiable lifestyle habits after breast cancer diagnosis, participation in sports activities is one of the factors that has been associated with reduced risk of mortality [7-9] as well as enhanced QoL [10-13]. A large body of short-term, structured exercise interventional programs have been completed among breast cancer patients. A few recently published meta-analyses have summarized these evidences, and reported positive associations between exercise and QoL.[14-16] The above-mentioned interventional studies were mainly based on Western population, while report from Asian population remains scarce. Another interesting finding was that supervised completely or partially training programs appeared to achieve greater improvement in QoL than those predominantly unsupervised or home-based ones.[17] On the other hand, observational studies evaluated patients’ self-motivated participation in sports activities, and have been reflective of patients’ daily habits. Up to now, several observational studies have been reported, including cohort studies and cross-sectional studies, suggesting an association between regular sports activities engagement and better QoL.[18-29] However, among these observational studies, only one cohort study have been reported in Chinese women with breast cancer.[20]” Please refer to Introduction section, Page 2, Line 53-68.

Reference

  1. Basha, M.A.; Aboelnour, N.H.; Alsharidah, A.S.; Kamel, F.H. Effect of exercise mode on physical function and quality of life in breast cancer-related lymphedema: a randomized trial. Supportive care in cancer : official journal of the Multinational Association of Supportive Care in Cancer 2021, 10.1007/s00520-021-06559-1, doi:10.1007/s00520-021-06559-1.
  2. Coletta, A.M.; Playdon, M.C.; Baron, K.G.; Wei, M.; Kelley, K.; Vaklavas, C.; Beck, A.; Buys, S.S.; Chipman, J.; Ulrich, C.M., et al. The association between time-of-day of habitual exercise training and changes in relevant cancer health outcomes among cancer survivors. PLoS One 2021, 16, e0258135, doi:10.1371/journal.pone.0258135.
  3. Smith-Turchyn, J.; McCowan, M.E.; O'Loughlin, E.; Fong, A.J.; McDonough, M.H.; Santa Mina, D.; Arbour-Nicitopoulos, K.P.; Trinh, L.; Jones, J.M.; Bender, J.L., et al. Connecting breast cancer survivors for exercise: protocol for a two-arm randomized controlled trial. BMC Sports Sci Med Rehabil 2021, 13, 128, doi:10.1186/s13102-021-00341-w.
  4. Tami-Maury, I.M.; Liao, Y.; Rangel, M.L.; Gatus, L.A.; Shinn, E.H.; Alexander, A.; Basen-Engquist, K. Active Living After Cancer: Adaptation and evaluation of a community-based physical activity program for minority and medically under-served breast cancer survivors. Cancer 2021, 10.1002/cncr.33904, doi:10.1002/cncr.33904.
  5. Lahart, I.M.; Metsios, G.S.; Nevill, A.M.; Carmichael, A.R. Physical activity for women with breast cancer after adjuvant therapy. The Cochrane database of systematic reviews 2018, 1, Cd011292, doi:10.1002/14651858.CD011292.pub2.
  6. Covington, K.R.; Hidde, M.C.; Pergolotti, M.; Leach, H.J. Community-based exercise programs for cancer survivors: a scoping review of practice-based evidence. Supportive care in cancer : official journal of the Multinational Association of Supportive Care in Cancer 2019, 27, 4435-4450, doi:10.1007/s00520-019-05022-6.
  7. Zhang, X.; Li, Y.; Liu, D. Effects of exercise on the quality of life in breast cancer patients: a systematic review of randomized controlled trials. Supportive care in cancer : official journal of the Multinational Association of Supportive Care in Cancer 2019, 27, 9-21, doi:10.1007/s00520-018-4363-2.
  8. Campbell, K.L.; Winters-Stone, K.M.; Wiskemann, J.; May, A.M.; Schwartz, A.L.; Courneya, K.S.; Zucker, D.S.; Matthews, C.E.; Ligibel, J.A.; Gerber, L.H., et al. Exercise Guidelines for Cancer Survivors: Consensus Statement from International Multidis-ciplinary Roundtable. Medicine and science in sports and exercise 2019, 51, 2375-2390, doi:10.1249/mss.0000000000002116.
  9. Blanchard, C.M.; Stein, K.D.; Baker, F.; Dent, M.F.; Denniston, M.M.; Courneya, K.S.; Nehl, E. Association between current life-style behaviors and health-related quality of life in breast, colorectal, and prostate cancer survivors. Psychology & Health 2004, 19, 1-13, doi:10.1080/08870440310001606507.
  10. Blanchard, C.M.; Courneya, K.S.; Stein, K.; American Cancer Society's, S.C.S., II. Cancer survivors' adherence to lifestyle be-havior recommendations and associations with health-related quality of life: results from the American Cancer Society's SCS-II. Journal of clinical oncology : official journal of the American Society of Clinical Oncology 2008, 26, 2198-2204, doi:10.1200/JCO.2007.14.6217.
  11. Chen, X.; Zheng, Y.; Zheng, W.; Gu, K.; Chen, Z.; Lu, W.; Shu, X.O. The effect of regular exercise on quality of life among breast cancer survivors. Am J Epidemiol 2009, 170, 854-862, doi:10.1093/aje/kwp209.
  12. Smith, A.W.; Alfano, C.M.; Reeve, B.B.; Irwin, M.L.; Bernstein, L.; Baumgartner, K.; Bowen, D.; McTiernan, A.; Bal-lard-Barbash, R. Race/ethnicity, physical activity, and quality of life in breast cancer survivors. Cancer epidemiology, bi-omarkers & prevention : a publication of the American Association for Cancer Research, cosponsored by the American Soci-ety of Preventive Oncology 2009, 18, 656-663, doi:10.1158/1055-9965.epi-08-0352.
  13. Blanchard, C.M.; Stein, K.; Courneya, K.S. Body mass index, physical activity, and health-related quality of life in cancer survivors. Medicine and science in sports and exercise 2010, 42, 665-671, doi:10.1249/MSS.0b013e3181bdc685.
  14. Voskuil, D.W.; van Nes, J.G.; Junggeburt, J.M.; van de Velde, C.J.; van Leeuwen, F.E.; de Haes, J.C. Maintenance of physical activity and body weight in relation to subsequent quality of life in postmenopausal breast cancer patients. Ann Oncol 2010, 21, 2094-2101, doi:10.1093/annonc/mdq151.
  15. Paxton, R.J.; Phillips, K.L.; Jones, L.A.; Chang, S.; Taylor, W.C.; Courneya, K.S.; Pierce, J.P. Associations among physical activi-ty, body mass index, and health-related quality of life by race/ethnicity in a diverse sample of breast cancer survivors. Can-cer 2012, 118, 4024-4031, doi:10.1002/cncr.27389.
  16. Gong, X.H.; Wang, J.W.; Li, J.; Chen, X.F.; Sun, L.; Yuan, Z.P.; Yu, J.M. Physical exercise, vegetable and fruit intake and health-related quality of life in Chinese breast cancer survivors: a cross-sectional study. Quality of life research : an interna-tional journal of quality of life aspects of treatment, care and rehabilitation 2017, 26, 1541-1550, doi:10.1007/s11136-017-1496-6.
  17. Shin, W.K.; Song, S.; Jung, S.Y.; Lee, E.; Kim, Z.; Moon, H.G.; Noh, D.Y.; Lee, J.E. The association between physical activity and health-related quality of life among breast cancer survivors. Health and quality of life outcomes 2017, 15, 132, doi:10.1186/s12955-017-0706-9.
  18. Hart, V.; Trentham-Dietz, A.; Berkman, A.; Fujii, M.; Veal, C.; Hampton, J.; Gangnon, R.E.; Newcomb, P.A.; Gilchrist, S.C.; Sprague, B.L. The association between post-diagnosis health behaviors and long-term quality of life in survivors of ductal carcinoma in situ: a population-based longitudinal cohort study. Quality of life research : an international journal of quality of life aspects of treatment, care and rehabilitation 2018, 10.1007/s11136-018-1807-6, doi:10.1007/s11136-018-1807-6.
  19. Ahn, S.J.; Kim, J.H.; Chun, M.; Yoon, W.S.; Rim, C.H.; Yang, D.S.; Lee, J.H.; Kim, K.; Kong, M.; Kim, S., et al. Physical activity status in relation to quality of life and dietary habits in breast cancer survivors: subset analyses of KROG 14-09 nationwide questionnaire study. Quality of life research : an international journal of quality of life aspects of treatment, care and rehabil-itation 2020, 29, 3353-3361, doi:10.1007/s11136-020-02585-4.
  20. Park, J.H.; Lee, D.H.; Kim, S.I.; Kim, N.K.; Jeon, J.Y. Moderate to vigorous physical activity participation associated with bet-ter quality of life among breast and colorectal cancer survivors in Korea. BMC cancer 2020, 20, 365, doi:10.1186/s12885-020-06819-z.

In addition, we have added the following description in the Introduction Section.

“Although numerous studies have reported positive associations between exercise and QoL among breast survivors, the underlying mechanisms remain unclear. Result from a lifestyle modification program, including adopting healthy diet pattern and doing exercises, suggested that lifestyle factors may impact positively on QoL possibly through weight loss.[30] It is well recognized that Asian women have lower body mass index (BMI) than Western women.[31] In addition, our previous study showed that less than 10% of Chinese women with breast cancer were obese according to the BMI classification of World Health Organization (WHO).[32] Taken together, these data suggested that the beneficial impact of exercise on QoL may differ between Western and Chinese women. Hence, studies that specifically address this aspect in Asian patients are still essential.” Please refer to Introduction section, Page 2, Line 69-78.

Reference

  1. Montagnese, C.; Porciello, G.; Vitale, S.; Palumbo, E.; Crispo, A.; Grimaldi, M.; Calabrese, I.; Pica, R.; Prete, M.; Falzone, L., et al. Quality of Life in Women Diagnosed with Breast Cancer after a 12-Month Treatment of Lifestyle Modifications. Nutrients 2020, 13, doi:10.3390/nu13010136.
  2. Chen, M.J.; Wu, W.Y.Y.; Yen, A.M.F.; Fann, J.C.Y.; Chen, S.L.S.; Chiu, S.Y.H.; Chen, H.H.; Chiou, S.T. Body mass index and breast cancer: analysis of a nation-wide population-based prospective cohort study on 1 393 985 Taiwanese women. International journal of obesity (2005) 2016, 40, 524-530, doi:10.1038/ijo.2015.205.
  3. Lei, Y.-Y.; Ho, S.C.; Kwok, C.; Cheng, A.; Cheung, K.L.; Lee, R.; Mo, F.; Yeo, W. Weight and waist-to-hip ratio change pattern during the first five years of survival: data from a longitudinal observational Chinese breast cancer cohort. BMC cancer 2021, 21, 839-839, doi:10.1186/s12885-021-08554-5.

Point 2: Results

3.1. Demographic, clinical and lifestyle characteristics of patients by level of sports activities at 18-, 36- and 60-month follow-up

This section should be rewritten as the result’s section reported below, reporting the specific data (mean value, p) without discussion.

Response 2: We thank the Reviewer for these comments. We have rewritten this section and reported the specific data (mean value or proportions, p) as follow.

“Data at 18-month follow-up are listed in Table 1. Compared to patients who did not participate in any sports activities, those who engaged in low- or high-level of sports activities were more likely to have college or above education (no vs. low-level vs. high-level sports activities group: 9.0% vs. 17.9% vs. 15.4%; P= 0.003). In addition, the proportions of patients who were post-menopausal at follow-up were significantly higher in those who were involved in low/high level of sports activities (74.9% vs. 77.5% vs. 82.8%; P= 0.027). The histology subtypes among the three groups differed slightly; more patients who had low/high sports activities had invasive ductal carcinoma (IDC) (79.8% vs. 85.7% vs. 85.1%; P= 0.043). Differences were also noted in the proportion of patients having HER 2 positive tumor (21.0% vs. 28.5% vs. 26.8%; P< 0.001) and who received chemotherapy (70.0% vs. 78.2% vs. 77.5%; P= 0.027). Of note, the proportion of patients with normal BMI was much higher in patients who participated in higher level of sports activities (41.9% vs. 42.3% vs. 53.0%; P= 0.001). Other demographic and clinical characteristics showed no significant difference between patients participating in different levels of sports activities.

Similar analyses were conducted at 36-month follow-up. The proportions of patients having no, low-level and high-level of sports activities were 29.4%, 38.9% and 31.6%, respectively. The significant differences observed-above in education level (patients who had college or above: 10.6% vs. 17.1% vs. 16.6%; P= 0.026) and menopausal status (patients who were post-menopausal: 75.5% vs. 77.6% vs. 83.1%; P= 0.039) were also noted at 36-month follow-up. In addition, patients who were involved in high-level of sports activities at 36-month follow-up tended to be older (mean age: 54.9 vs. 54.7 vs. 56.4; P= 0.019) and had higher household income (patients with household income ≥30,000 HKD/month: 17.2% vs. 22.8% vs. 28.1%; P= 0.009) than those who did not engage in any sports activities. Moreover, patients who had low-level of sports activities at 36-month follow-up were more likely to have endocrine therapy (71.0% vs. 78.8% vs. 71.3%; P= 0.018) and report menopausal symptoms (60.7% vs. 70.1% vs. 63.2%; P= 0.017), than those who did not engage in any sports activities.

At 60-month follow-up, the proportions of patients who belonged to no, low-level and high-level of sports activities were similar to those during the 36-month follow-up, with the corresponding figures being 32.8%, 36.7% and 30.5%, respectively. When compared to patients who did not engage in any sports activity, those who were engaged in low- or high-level of sports activities were more likely to have college or above education (10.7% vs. 19.0% vs. 17.4%; P= 0.004). In addition, the proportion of patients with normal BMI were higher in those who were engaged in low- and high-level of sports activities (37.4% vs. 38.8% vs. 41.8%; P= 0.001). No significant differences were found regarding other characteristics.” Please refer to Results section, Page 5, Line 196-231.

Point 3: Discussion

Please comment these results of the 3.1 section.

Response 3: We thank the Reviewer for these comments. We have added a paragraph to discuss the results of the 3.1 section as follow.

“This study compared the characteristics of patients who participated in different level of sports activities. Of note, patients who attended college or above were more likely to involve in higher level of sports activities at all the three follow-ups. This phenomenon has also been reported in a previous study.[20] This study also showed that higher proportion of women who were post-menopausal participated in higher level of sports activities, suggesting that might have more spare time that allowed them to focus more on their lifestyle habits. They may also be seeking potential maneuvers, such as sports activities, to relief menopausal symptoms. Moreover, patients with normal BMI were associated with higher level of sports activities participation. Patients with normal BMI might be more health-conscious; on the other hand, normal BMI could be the result of regular sports activities.” Please refer to Discussion section, Line 336-346.

Reference

  1. Chen, X.; Zheng, Y.; Zheng, W.; Gu, K.; Chen, Z.; Lu, W.; Shu, X.O. The effect of regular exercise on quality of life among breast cancer survivors. Am J Epidemiol 2009, 170, 854-862, doi:10.1093/aje/kwp209.

Point 4: Discussion

Please evidence if your data differ from those reported in manuscripts on western breast cancer survivors.

Response 4: We thank the Reviewer for these comments. We have added some discussion as follow.

“Similar to previous studies, the present study provided important data from an Asian population, supporting that regular engagement in sports activities, especially that could meet the recommended exercise level, were associated with better QoL in several items. Of note, the questionnaires used to measure QoL in earlier studies were not identical. None the less, the significant associations between higher level of sports activities participation and global health status, physical functioning, role functioning as well as emotional functioning in the present study were consistent with that reported in US population.[21] However, the significant association between higher level of sports activities participation and social functioning in the US population were not observed in the present study.” Please refer to Discussion section, Line 396-404.

Reference

  1. Smith, A.W.; Alfano, C.M.; Reeve, B.B.; Irwin, M.L.; Bernstein, L.; Baumgartner, K.; Bowen, D.; McTiernan, A.; Ballard-Barbash, R. Race/ethnicity, physical activity, and quality of life in breast cancer survivors. Cancer epidemiology, biomarkers & prevention : a publication of the American Association for Cancer Research, cosponsored by the American Society of Preventive Oncology 2009, 18, 656-663, doi:10.1158/1055-9965.epi-08-0352.

Point 5:

References

Please update this reference:

World Cancer Research Fund/American Institute for Cancer Research. Food, Nutrition, Physical Activity, and the Prevention of Cancer: A Global Perspective. Washington, DC: AICR, 2007.

Response 5: We thank the Reviewer for the kind reminder. We have added update this reference as follow.

Reference 42. World Cancer Research Fund/American Institute for Cancer Research Diet, Nutrition, Physical Activity and Cancer: A Global Perspective. Continuous Update Project Expert Report 2018. [(accessed on 11 November 2021)]; Available online: https://www.wcrf.org/diet-and-cancer/. Please refer to Reference section, Line 568-570.

Round 2

Reviewer 2 Report

The authors answered my comments adequately and appropriately.  One final comment is to consider defining "sports activities".  Is this equivalent to moderate to vigorous physical activity? if so, consider changing to avoid confusion.

Author Response

Point 1: Comments and Suggestions for Authors

The authors answered my comments adequately and appropriately.  One final comment is to consider defining "sports activities".  Is this equivalent to moderate to vigorous physical activity? if so, consider changing to avoid confusion.

Response 1: We thank the Reviewer for the kind reminder. Sports activities means physical activity in doing sports. We did not include other physical activities like housework, childcaring etc. Although we used the modified Chinese Baecke questionnaire to measure patients’ daily participation of physical activity, which included physical activity at work, in doing housework, at leisure time and in doing sports, the present study only used one part of the data, namely that of doing sports. We have now clarified this is the amended manuscript.

“Sports activities means physical activity in doing sports. We did not include other physical activities like housework, childcaring etc.” Please refer to Methods section, page 3, Line 123-124.

Please also kindly note that we considered the inclusion of sports activities instead of all physical activities in our analysis as a limitation of our study, and this had been mentioned in our manuscript. Please refer to Discussion Section, Line 446-448.

Reviewer 3 Report

All the comments have all been taken into account and the paper has been changed correctly. The introduction was expanded as required and the results and discussions reworked appropriately. The bibliography has also been updated. Congratulations on the excellent work.

Author Response

Response 1: We thank the Reviewer for these comments.